# Cell Wall Composition of Hemp Shiv Determined by Physical and Chemical Approaches

**DOI:** 10.3390/molecules26216334

**Published:** 2021-10-20

**Authors:** Maya-Sétan Diakité, Hélène Lenormand, Vincent Lequart, Santiago Arufe, Patrick Martin, Nathalie Leblanc

**Affiliations:** 1UniLaSalle, Université d’Artois, ULR7519—Transformations & Agro-Ressources, Normandie Université, F-76130 Mont-Saint Aignan, France; maya.diakite@unilasalle.fr (M.-S.D.); santiago.arufe@gmail.com (S.A.); nathalie.leblanc@unilasalle.fr (N.L.); 2Université d’Artois, UniLaSalle, ULR7519—Transformations & Agro-Ressources, F-62400 Béthune, France; vincent.lequart@univ-artois.fr (V.L.); patrick.martin@univ-artois.fr (P.M.)

**Keywords:** biomass, hemp shiv, cell wall, Van Soest method, TGA, bio-based materials

## Abstract

The use of agricultural by-products in the building engineering realm has led to an increase in insulation characteristics of biobased materials and a decrease in environmental impact. The understanding of cell wall structure is possible by the study of interactions of chemical compounds, themselves determined by common techniques like Van Soest (VS). In this study, a global method is investigated to characterise the cell wall of hemp shiv. The cell wall molecules were, at first, isolated by fractionation of biomass and then analysed by physical and chemical analysis (Thermal Gravimetric Analysis, Elementary Analysis, Dynamic Sorption Vapor and Infra-Red). This global method is an experimental way to characterise plant cell wall molecules of fractions by Thermal Gravimetric Analysis following by a mathematical method to have a detailed estimation of the cell wall composition and the interactions between plant macromolecules. The analyzed hemp shiv presents proportions of 2.5 ± 0.6% of water, 4.4 ± 0.2% of pectins, 42.6 ± 1.0% (Hemicellulose–Cellulose), 18.4 ± 1.6% (Cellulose–Hemicellulose), 29.0 ± 0.8% (Lignin–Cellulose) and 2.0 ± 0.4% of linked lignin.

## 1. Introduction

For thousands of years, lignocellulosic biomass like hemp has been cultivated. At the beginning, crop production was based on fibers to produce garments or ropes for the shipbuilding sector. Later on, culture was extended to produce animal food, new high molecules for the pharmaceutical sector, second-generation biofuels and more recently waste has been used as bio-based materials for building engineering. These kinds of bio-based products are interesting due to their insulation capacities, low carbon impact or their ability to increase air quality. These products contribute to increase local employment (agriculture or agronomy) and decrease the impact of carbon linked to importations. Thus, over recent decades, research topics have focused on plants valorisation into bio-based building materials like particle boards or mortars. The technical interest to incorporate biomass into building materials lies in the contribution of plant particles to hygrothermal [1], mechanical [2] and acoustic [3] performances of materials. There is much literature on these aspects and because biomass does not only bring positive functionalities on bio-based materials, studies about the properties of biomass were investigated. Two of the most important parameters for bio-based building materials are microstructure and biochemical composition. Microstructure defines the quantity and type of porosities [4,5]. Biochemical composition may bring positive aspects, e.g., in the case of self-adhesion capacity in the case of binderless particleboards [6], but may also cause troubles in the quality of bio-based materials [7], e.g., in the case of disturbances in setting time of mortars [8] due to solubilization of molecules from biomass [9]. Hemp is the usual biomass in bio-based building materials but today some plant particles are being investigated [10,11]. As plant matter is a natural matter in contrast with conventional building materials submitted to rules and international standards, the variability of plant particles properties from biomass must be studied. The most usual technique for the determination of biochemical composition is the Van Soest method [12]. Table 1 summarizes some results of hemp shiv and highlights a relative variability. Indeed, environmental parameters, conditions, species or retting can influence biochemical composition [8].

Other global cell wall characterization techniques like Prosky, Uppsala, non-starch polysaccharides or Selvendran, exist to analyze cell wall biomass [21]. Each technique gives the opportunity to bring specific information but results from different techniques are hardly comparable and do not bring information about chemical interactions between plant cell wall molecules.

Nowadays, main macromolecules from biomass are well categorised [22] (i.e., cellulose, pectin, protein, hemicellulose and lignin) but, the structure and arrangement of cell wall’s complex network remain misunderstood [23,24]. Within a lignocellulosic plant matrix, hemicellulose, lignin, and cellulose molecules interact to form a rigid molecular network called “Lignin Carbohydrate Complex” or LCC [23,24]. LCC arrangements form complex structures in lignocellulosic biomass depending on wood type. Softwood or gymnosperms wood is composed of 33–42% cellulose, 22–40% hemicellulose, 27–32% lignin and 2–3.5% extractive. Hardwood or angiosperms wood contain around 38–51% cellulose, 17–38% hemicellulose, 21–31% lignin and 3% extractives [25].

This difference in cell wall composition between wood types can be also observed in the same molecular families [24]. Indeed, xylan and glucomannan form the basic backbone polymer of hemicellulose wood. In hardwood like hemp, xylan is the main compound while glucomannan is predominant in softwood [24]. In hardwoods, xylans were present in *O*-acetyl-*O*-methylglucuronoxylan, with one *α*-(1-2)-linked 4-*O*-methyl glucuronic acid substituted by 10 to 20 *β*-(1-4)-d-xylopyranose units [26]. The heteropolymer, lignin composed of coumaryl (H), coniferyl (G) and synapyl (S) alcohol in *trans* configuration association is extremely associated with hemicellulose by benzyl ether or ester linkages [26,27].

The proportion of each molecule depends on plant type. Subunit G and S are essentially in Dicotyledon species like hemp. Lignin and hemicellulose were also meant to be linked to cellulose (*β*-*O*-glucose) by hydrogen bonds [22,25].

These intermolecular interactions contribute to the strengthening of the plant matrix and constitute a lock to the extraction of parietal compounds. An overview of plant cell wall composition was proposed in Viel et al. [10] based on literature to imagine the intermolecular interactions between the different families of macromolecules.

Plant cell wall interaction remains a challenge. In this way, a deeper understanding of plant composition is required to understand the interactions of molecules released by plant particles during the manufacturing process of bio-based materials.

This work proposes a global way to characterise plant cell wall structure, especially in hemp shiv, with two approaches:Chemical fractionation of the biomass to isolate cell wall molecules, by a non-conventional and non-destructive Van Soest version (not the final step of calcination).A physical and chemical characterisation of fractions from fractionation called “Van Soest Fractions” by Thermal Gravimetric Analysis (TGA) under argon or oxygen, Elementary Analysis (EA), Fourier Transformed Infrared (FT-IR) and Dynamic Vapor Sorption (DVS).

The results obtained from this global way were analysed and compared to the conventional Van Soest method.

## 2. Materials and Methods

### 2.1. Preparation of Hemp Shiv

The variety of hemp was FEDORA 17. Hemp shivs were produced in Normandy Region in France by defibration of hemp stems in a local producer’s transformation unit. The particles of hemp shiv were sieved to remove all residual fibers. Then, particles were crushed into powder using a mill with a mill-sieve of 1 mm if the procedure requested it and plant matter was stored at 40 °C.

### 2.2. Van Soest Method on Raw Hemp Shiv

Van Soest method (VS) [12] is a global cell wall quantification used to quantify by gravimetric method all cell wall compounds like cellulose, hemicellulose, lignin, soluble compounds and ashes. Van Soest method was performed with a FibertecTM 8000 semi-automatic machine. Solvents used during the process are Neutral Detergent Fiber (NDF, VWR, Chemicals, 305320.5000) to remove soluble compounds (pectins, oils, ashes, water, sugars, etc.), Acid Detergent Fiber (ADF, VWR Chemicals, 305319.5000) to remove hemicellulose and sulfuric acid (H_2_SO_4_ 72%, Carlo Erba reagents, 502771) to remove cellulose. Then, sulfuric acid treatment was followed by neutralization with deionized water.

After each treatment, samples are dried at 105 °C for 16 h, weighed and dried in an oven at 480 °C for 6 h to remove lignin and obtain ashes. For the analysis, six repetitions were performed.

### 2.3. Non-Destructive Van Soest Method and Preparation of Fractions

The non-destructive version of the Van Soest [12] method (NSVS), based on original VS, was applied on hemp shiv particles and powders. Original vs. counts three chemical steps. In NDVS, samples (fractions) were collected after each step treatment (NDF, ADF and H_2_SO_4_). Then, samples were dried at 105 °C for 16 h. Calcination was not carried out. That is why the method is called “non-destructive”. Schema of NDVS is proposed in Figure 1. Three repetitions were performed.

Theoretically [12,21] the first fraction (Fraction A (FA)) contains cellulose, hemicellulose, lignin and ashes; the second (Fraction B (FB)) contains cellulose, lignin and ashes and the last (Fraction C (FC)) contains lignin and ashes. Samples are stored in plastic jars at room temperature.

Milled samples, including control (hemp shiv without treatment) and FA, FB and FC were analysed: hemp shiv powder treated by NDVS was used for TGA, Elementary analysis and Infrared, and hemp shiv particles treated by NDVS for DVS (Figure 1).

### 2.4. Thermal Gravimetric Analysis (TGA)

The experiments were carried out on a Netzsch TG209 F1 machine. The samples (three repetitions) were heated 30 °C to 600 °C at the rate of 10 °C/min. Slow pyrolysis was performed under two atmospheres: argon and oxygen, both have been employed independently. The mass recorded were converted to percentage mass losses to have normalized data. The mass variation as a function of temperature was recorded. Mass loss percentage (m) peaks were identified with a determination of onset, peak and final temperatures of thermal transitions in first and second derivatives, Table 2 [28]. The deconvolution of each whole curve of degradation provides access to chemical composition [29,30]. Due tue the complexity of cell wall compounds, mathematical data treatment started with FC first and second derivative, following by FB, FA and Control.

### 2.5. Fourier Transformation Infrared Spectroscopy (FT-IR)

Infrared determines the molecular composition of samples based on the type of link between molecules. Without advance preparation, samples were placed on a crystal and the experiment was carried out on CARY 630 FTIR spectrometer machine with MicroLab Agilent Technologies Software. Spectra were obtained after 4 scans in a spectral range of 800–4000 cm^−1^ for each sample.

### 2.6. Elemental Analysis (EA)

Elemental analysis is a destructive analytical technique that can be qualitative and/or quantitative and is used to determine the elementary composition of a sample or its purity that can be qualitative and/or quantitative.

The samples placed in a capsule are degraded by pyrolysis under excess oxygen and the combustion products (CO_2_, H_2_O and NO) are recovered. The masses of the combustion products are then used to calculate the elementary composition of the sample. Analysis provides access to the carbon, hydrogen, nitrogen, sulfur rate. Oxygen rate is determined by subtracting the rates of other elements present. Three repetitions were performed.

In this way protein levels through the nitrogen level can be obtained [16,21]. Proteins were calculated with Equation (1):% Proteins (g_proteins_. gN^−1^) = % Nitrogen (g) × 6.25(1)

Carbon, Hydrogen, Nitrogen and Sulfur (CHNS) determination was carried out in four repetitions and the analysis was carried out simultaneously by the combustion analyser. The machine used for the analysis was the Elementar Vario EL cube.

### 2.7. Dynamic Vapour Sorption (DVS)

DVS is a gravimetric method that consists of measuring the sample’s mass variation by applying a water relative humidity (*φ*) protocol changing from 0% to 90% at 23 °C. Different equilibrium points were determined corresponding to a fixed *φ* 0%, 5%, 10%, 15%, 20%, 30%, 35%, 50%, 75%, 85% and 90% for adsorption experiments and 0%, 15%, 30%, 35%, 50%, 75%, 85%, 90% for desorption experiments. Every point was determined in duplicate. The machine software was Sorption Test System from Prolumid.

Adsorption and desorption isotherms got three areas and each area corresponds to adsorption state (monolayer, multilayers, and liquid water). Few models exist to predict physical water adsorption, among them Brunauer Emmet Teller model [31] (Equation (2)) is the commoner used to explain monolayer at the beginning of adsorption (from 0% to 35% of *φ*) characteristic of Van der Waals forces between water and hydrophiles groups.
(2)$$X_{eq}=\frac{X_{m}C\phi }{\left(1-\phi \right)\left(1+\left(C-1\right)\phi \right)}$$

This model was employed to fit water sorption experimental data (equilibrium moisture content, *X_eq_*, and relative humidity, *φ*) and then obtain the specific surface area, *a_s_* (m^2^/g) [31,32] (Equation (3)).
(3)$$a_{s}=\frac{X_{m}La_{m}}{M_{w}}$$
where *X_m_* is the monolayer moisture content (kg water/kg dried solid, d.b.) and *C* (dimensionless). BET was applied assuming that *X_m_* of BET model represents the quantity of water molecules that cover the entire surface as a monolayer and that no diffusion of water through the material has taken place.

Where *a_s_* is the specific surface area of the sample (m^2^/g), *L* the Avogadro constant (6.023 × 10^−23^ mol^−1^), *a_m_* cross-sectional area of water molecule (1.08 × 10^−19^ m^2^) and *M_w_* molecular weight of water molecule (18 g·mol^−1^).

## 3. Results and Discussions

### 3.1. Chemical Characterisations

#### 3.1.1. Conventional Van Soest Method

Conventional Van Soest method was applied on raw hemp shiv to determine proportions of molecules families [12]. Theoretically, as the vs. steps are taken, molecules are specifically eliminated. Results, presented in Table 3, show that the organic part of hemp shiv contains 50.4% of cellulose, 22.2% of hemicellulose, 19.1% of soluble compounds (pectins, lipids, sugars, ashes, proteins, etc.) and 8.3% of lignin.

All these molecules represent 97.2% (*w*/*w*) of organic mass, and the rest is the inorganic matter (also called ashes). Even if the literature highlights a high variability in Van Soest results, our results agree with the bibliography and present a high reproducibility due to standard deviation lower than 8% for all cell wall compounds. According to literature (Table 1), hemp shiv contains around 44.0–51.6% cellulose, 6.4–27.0% hemicellulose, 8.0–28.0% lignin, 1.0–29.4% soluble compounds and 1.2–8.8% ashes. The high variability observed in proportions of molecules families can be explained by some differences in growing conditions like pluviometry [10], weather or temperature [10,15] which have an impact on chemical composition. Retting was also described to have a huge impact; it seems to be a major parameter to cell wall composition of hemp shiv. In Arufe et al. [18] study, retting was performed during periods of 0 to 62 days. Soluble compounds decrease from 21.1% to 12.3%, hemicellulose from 21.5% to 11.9%, cellulose from 46.1% to 39.4%, lignin from 8.5% to 7.5% and ash from 2.8% to 1.3%. The more retting there is the lower cell wall compounds, especially hemicellulose, soluble compounds, cellulose, and ash rate. Mass loss during retting is due to microorganism presence like fungi or bacteria which produce specific enzymes like cellulase, hemicellulases (xylanase, xylosidase, mannanase, arabinose, etc.) pectinase (polygalacturonase, pectine methylesterases or acetylesterases, etc.) or ligninases (phenol oxidase or peroxidase) product by fungi which disorganize cell wall structure by destroying covalent bonds into LCC complex [33]. During the beginning of retting, hemicellulose and soluble compounds were the first to be removed from the sample. Thus, hemicellulase and pectinase seem to be the most produced enzymes by a microorganism that they use in their metabolism. However, the literature suggests that lignin structure protects biomass from microorganism degradation. Thus, naturally, a low lignin rate facilitates retting, and this could explain the low degradation of lignin which is less required to eliminate allowing sugar accessibility in the sample [33,34]. Results from Table 1 also highlight the huge fluctuation of lignin rate (from 8.0 to 28.0%) compared to other cell wall compounds. The hemp variety could explain lignin rate variation, and therefore the hemp’s ability to ret since plant botanic species would have an impact on lignin biosynthesis, but no study really confirms this hypothesis [15].

Drying of hemp shiv for 16 h under 105 °C generates 3.4% mass loss. This mass loss percentage is linked to free water adsorption at room temperature. So, hemp shiv was not entirely dried and still contains bound water, which participates in soluble compounds quantity. Soluble compounds represent several types of molecules including water. According to Van Soest results, 18.6% of dry mass correspond to soluble compounds in neutral detergent including at least 3.4% free water (Table 3).

#### 3.1.2. Elementary Analysis

Elementary Analysis was performed on crushed samples of hemp shiv. Sample “Control” corresponds to raw hemp shiv and FA, FB and FC correspond to samples obtained by the non-destructive Van Soest method. Results are given in Table 4 and Figure 2. Protein rate based on nitrogen percentage was calculated with Equation (1). Control sample contains 0.01% nitrogen and so less than 0.1% proteins. Each fraction, except FC, proposes a similar proportion between the different elements (Figure 2, Table 4). From control to FB, the sample contains around 45% carbon, 6% hydrogen, 0.01% nitrogen and 0.01% sulfur and 48% calculated oxygen. In these three samples, cell wall compounds were removed but no real elemental differences were observed. FC shows lower carbon, hydrogen, and oxygen rates of 53.3%, 5.4% and 39.4%, respectively, but the higher sulfur rate (1.86%) and no change for nitrogen rate (0.03%). All samples seem to present high reproducibility according to low standard deviation. Control results seem to agree with the bibliography with a similar element rate. Indeed Brazdausks et al. [35] studied elemental analysis on the hemp shiv part of “Bialobrzeskie” variety. Results were 47.4% carbon, 5.3% hydrogen, 0.6% nitrogen, 0.2% sulfur and 46.8% oxygen obtained by difference.

For all samples, low nitrogen rate could be explained by the secondary cell wall implementation. Indeed, after the implementation hemp shiv will be devitalised, but some proteins like arabinogalactans proteins (AGP) which are extremely linked to the cell wall can be there in low proportion [36].

In comparison to control, FC carbon presents an increased carbon rate of 17.7% and a decreased hydrogen rate of 12.9%. The nitrogen rate did not significantly change. The sulfur rate increased from 0 to 1.86% and oxygen percentage decreased by 18.4%. In FC, the increase of carbon rate is related to the dominant presence of lignin heteropolymer. Lignin, corresponding to 8.1% of dry matter (Table 3) in Dicotyledons species like hemp are mostly composed of S and G subunits which are easily oxidable [37]. During the process, FC was oxidized into by-products like CO_2_ during the combustion process. Raising sulfur is due to H_2_SO_4_ treatment that leaves some traces of sulfur. Hydrogen rate and especially calculated oxygen rate decreased because of cellulose elimination mainly composed of hydroxyl groups and corresponding to 49% of dry mass (Table 4, Figure 2) [35,36,37,38].

#### 3.1.3. Fourier Transformed InfraRed Spectroscopy (FT-IR)

Figure 3 presents the IR spectra of control and fractions samples.

Results of fractions FTIR are in accordance with the literature [8,10,39,40]. Typical peaks of specific chemical functions in hemp shiv powder are present. For the control sample, the first absorption band present at 3336 cm^−1^ represents O-H stretching of lignin and polysaccharides. This band is followed by a peak at 2918 cm^−1^ corresponding to the C-H bond of methyl, methylene or methane from cellulose, hemicellulose, polysaccharides, and wax. At 2891 cm^−1^, the stretching peak is associated with CH_2_ asymmetrical and symmetrical stretching from fats, lignin or polysaccharides. The 1736 cm^−1^ stretching peak represents C=O unconjugated bond associated with acetyl groups from lignin and hemicellulose xylans. The 1597 cm^−1^ stretching peak corresponds to C=C lignin bond. The 1321 cm^−1^ stretching peak is a C=O bond of syringyl alcohol subunit from lignin and cellulose. The 1236 cm^−1^ stretching peak represents the C-O bond of the aromatic ring of lignin, and the last stretching peak at 1031 cm^−1^ corresponds to C-C, C-OH, C-H of ring and side group of hemicellulose, cellulose, lignin and pectins (Figure 3). Results are quite similar between the control, FA and B, but slightly different with FC. FC shows a lower intensity of peaks and a supplementary peak between 1500–1000 cm^−1^, suggesting the presence of bonds between lignin and ether, ester functions [10]. After the last treatment allowing FC, the large peak of low intensity at 1031 cm^−1^ reveals the presence of remaining cell wall compounds.

FT-IR results, especially in the control, agree with the literature, but other fractions corroborate the elimination of cell wall compounds by treatment vs. steps, without quantitative information given by the Van Soest method but with information about interactions between molecules.

### 3.2. Physical Characterisations

#### 3.2.1. Thermal Gravimetric Analysis (TGA)

The non-destructive version of the Van Soest method was carried out to generate crushed samples with a specific composition. All generated fractions were analysed by TGA. The mass loss curves as a function of temperature for the different fractions (FA, FB and FC) of hemp shiv and for raw hemp shiv (control) are shown in Figure 4a (TGA under argon) and Figure 4b (TGA under oxygen).

In wood combustion, reactions start by pyrolysis between 200–700 °C in inert conditions and produce volatile molecules (e.g., alcohol, H_2_, CO, CH_4_) and charcoal, to induce wood destructuration. The presence of volatile compounds produced by pyrolysis lead to a combustion reaction and took place under oxygen at 750 °C. During this step, volatile compounds interact with oxygen until ignition.

Mass losses under argon (Figure 4a) show that real degradation starts at around 200 °C. Between 200 °C and 330 °C, control is slightly more degraded followed by FA, FB and FC. From 330 °C to 600 °C, the tendency changes and FB is the most degraded, followed by FA, control, and FC. Moreover, the slope of curves increases between control, FA and FB. The degradation was uncompleted due to the use of argon avoiding samples oxidation. Except for FC, the higher the number of treatment steps, the more degradation is achieved and the purer the degraded sample was. As the NDVS steps are taken, compounds were eliminated. The successive treatments make specific cell wall compounds accessible. The major mass loss is observed between 200–400 °C for all fractions, range of degradation temperatures of pectins, hemicellulose, cellulose in increasing order of degradation temperature. As FC is less degraded in a large range of temperatures than other samples, FC seems to be more stable at high temperatures than other samples, thanks to the high presence of lignin [29,41,42]. At the end of this analysis (600 °C), the most degraded sample is FB with 88% of mass loss, followed by FA with 81%, control with 78% and FC with only 32%.

TGA curves under oxygen (Figure 4b) show different behaviour. Degradation starts at around 150 °C. The control sample presents the lowest temperature of beginning degradation followed by FA, FB and FC. Thus, between 250–550 °C degradation curve behaviour changes. Control and FA, FB curves are clearly defined without superposition and have the same allure with two jumps: the first one, the highest at around 280 °C showing an important slope and the second one at around 350 °C. TGA analyses under O_2_ demonstrate the good fractionation of vs. method. Family compounds are selectively eliminated by the different steps of VS. Results are in accordance with literature: when temperature increases, the first family of molecules eliminated is pectins, followed by hemicellulose, cellulose and lignin. The difference in the height of the jumps (the first one being higher than the second one) corroborates the major proportion of polysaccharides in the samples’ jump. According to Van Soest results, hemicellulose, and cellulose (which are degraded between 200–400 °C) represent around 70% of total biomass. This percentage correlates with the important mass loss in the control, FA and FB during this range. In Dicotyledons, the primary cell wall contains 20–25% xyloglucans and 20–30% glucuronoxylans in the secondary cell wall [43]. However, the vascular plant has arabinoxylans or glucomannans in different proportions. These kinds of molecules could be degraded during the reaction. The second jump corresponds to the lignin degradation. The progressive removal of some compounds improves the access to other compounds. FB is theoretically composed of cellulose, lignin and ashes whereas FC is only by lignin and ashes. Comparison between FB and FC curves suggests two hypotheses: At first, hemicellulose was partially removed from the sample as CO_2_ and char, where xylans are partially depolymerized from the sample [42,44,45,46]. Contrary to the FB curve, the FC curve presents only one jump, which is progressive, suggesting that cellulose decrystallisation and elimination make accessible complexes (Lignin–Hemicellulose) with strong bonds. In consequence, this allows making reducing functions and alcohol functions of lignin accessible to produce volatile and oxidative by-products [39,44,47]. At the end of the analyse (600 °C), 100% of all samples are degraded.

The greatest interest of TGA analysis was to quantify cell wall compounds by calculation of mass losses during the decomposition temperature range of cell wall molecules. The composition of the samples is obtained by deconvolution of derivatives. Curves showed peaks corresponding to the degradation of specific compounds during a specific temperature range. Sometimes, peaks were wide and/or showed shoulders, which implied the presence of strong bonds or complexes between macromolecules. Analysis of TGA derivatives curves has been progressive, starting with the FC sample to determine the temperatures range, then with the FB sample and so on. Deconvolution of derivatives curves (first and second) made a possible estimation of samples composition. Curve ratios of the first derivative between fractions (FA or FB or FC) and Control are presented in Figure 5, in order to find specific ranges of degradation for each fraction. Data was removed from 400 °C to 600 °C because of auto combustion behaviours of Van Soest fractions under oxygen which distorts mass loss derivatives.

The composition of samples obtained by analysis of TGA with inert conditions (under Ar) are presented in Table 5. Control sample is composed of 2.7% water, 3.5% pectins, 24.7% complexes (Hemicellulose–Cellulose) and 38.8% complexes (Lignin–Cellulose) into control sample. These results represent 70% of biomass decomposition. Similar percentages are found for FA, with 2.0% pectins, 23.5% complexes (Hemicellulose–Cellulose) and 53.7% complexes (Lignin–Cellulose). FB shows 1.2% water and 83.8% cellulose, pectins and hemicellulose linked. FC shows 2.3% water, 4.0% pectins, 8.6% complexes (Hemicellulose–Cellulose), 17.6% complexes (Lignin–Cellulose). According to the total percentage under argon (70%), reactions are limited to pyrolysis and limit the yield of degradation. Results also suggest that all cell wall molecules are not removed by Van Soest treatments. In comparison to other samples, FC is the least degraded, because the theoretical major compound of FC is lignin. The FC degradation is uncompleted and makes low mass loss degradation (32%). Moreover, the lignin structure gives thermal stability to FC under a large range of temperature as compared to other fractions. Also, calculations with first and second derivatives of FC suggest that some cell wall compounds are still present there, which also could be represented on mass loss in FC fraction (32%).

The composition of samples obtained by analysis of TGA with oxidative conditions (under O_2_) are presented in Table 5. The control sample contains 2.5% water, 4.4% pectins, 42.6% complexes (Hemicellulose–Cellulose), 18.4% complexes (Cellulose–Hemicellulose), 29.0% complexes (Lignin–Cellulose) and 2.0% lignin for control sample. (Cellulose–Hemicellulose) contain more cellulose than (Hemicellulose–Cellulose). Total mass decomposition represents 100% of the biomass. FA and FB obtained the same order of the magnitude for water, linked hemicellulose and linked lignin percentages. FC contain 3% water, 5.2% pectins, 19.7% (Hemicellulose–Cellulose), 26.4% (Cellulose–Hemicellulose) and 48.3% (Lignin–Cellulose). These total percentages could be explained by the complete combustion under an oxygen atmosphere which allows pyrolysis and combustion reactions (Table 5). According to the literature, xylan pyrolysis is quite similar to that of cellulose because of structure similitudes [45,46]. Because of its complexity and abundant side chains attached to the hemicellulose backbone, xylan degradation is complex [43]. Volatile compounds like 1-hydroxy-2-butanone, 4-hydroxy-5,6-dihydro-(2*H*)-pyran-2-one, 1-hydroxy-2-propanone (acetol), acetaldehyde or hydroxyacetaldehyde (HAA) are widely produced during pyrolysis in high temperatures. Some models like the Broido-Shafizadeh model (B-S model) describe cellulose chemical pyrolysis [46]. The reaction starts with cellulose activation followed by several depolymerisation by volatile compounds production like Levoglucosan (LG), char and gases [42,44]. In investigated hemp shiv, cellulose represents 49% of cell wall.

Treatment of TGA curves allowed an accurate estimation of cell wall composition. Globally, TGA analysis revealed the presence of complexes and results from TGA under oxygen are more informative than those obtained under argon. Indeed, the composition is more detailed with the presence of complexes (Cellulose–Hemicellulose).

#### 3.2.2. Dynamic Sorption Vapour (DVS)

Control and fractions generated by NDVS method were analysed by DVS. Figure 6 shows curves of equilibrium moisture content (*X_eq_*) given by Equation (2), as a function of relative humidity. The high reproducibility of this technique is noticed with a maximum standard deviation of 4%.

Between 0% and 50% relative humidity, FA, FB and FC and control have similar behaviour. This trend changes at relative humidity higher than 50% HR. For example, at 90% relative humidity, FB has the lowest equilibrium moisture content (*X_eq_*) with 18%, followed by FA with 24%, control with 28% and FC with 32% (Figure 6). All samples’ curves show a hysteresis with higher equilibrium moisture contents on desorption than on sorption.

The literature presents three types of water: constitutive water which is extremely linked to the cell wall and so hard to remove; linked water which is water to the cell wall by hydrogen bonds and free water, stored in the lumen by capillary forces. According to IUPAC classification, hemp shiv has type II-isotherm curves which represent adsorption behaviour of microporous and non-porous adsorbent [48,49]. Some studies found moisture sorption could be affected by many factors like wood composition, especially hemicellulose, cellulose, lignin, and extractives. The presence of rich hydroxyl groups in cellulose and hemicellulose makes a great contribution to the hygroscopicity of wood, while lignin is a relative hydrophobic heteropolymer [32]. Thus, these results reveal the importance of decrystallised hemicellulose and highlight hygroscopic behaviour of xylans in hemp shiv.

DVS adsorption behaviour results are mainly explained by the chemical composition of samples. However, as the used chemical detergents of Van Soest protocol disorganise the plant matter, the porosity of samples and so the sensibility towards water were modified (Figure 6). Removal of, firstly, soluble compounds (case of FA) and then, soluble compounds and hemicellulose (case of FB) from the sample decreases the ability of the remaining matter to interact with water. The lower the hydrophilic groups like OH- from hemicellulose or pectins are, the lower *X_eq_* is. When soluble compounds, hemicellulose and a part of cellulose are removed from the sample (case of FC), the ability of the remaining matter to interact with water is high. So, FC is composed of hygroscopic molecules. According to conclusions from TGA results (Table 5), FC is still composed of a high proportion of cellulose. In addition to partial removing cellulose, H_2_SO_4_ treatment increases access to water to decrystallised cellulose and lignin molecules and promotes interactions between OH- functions and free-water [46,50,51].

BET method was employed to define monolayer water adsorption [31]. Results are presented in Figure 7. The specific area decreases with the number of treatment steps (Control, FA and FB) except for FC which is slightly higher than FB. Specific area (Equation (3)) of the control, FA, FB and FC are 157.3 ± 7.7, 147.0 ± 4.8, 94.3 ± 3.7 and 119.5 ± 1.7 m^2^/g, respectively (Figure 7).

As FA is free of soluble compounds and a_s_ modification between control and FA is not significant (about 6.5% of reduction), soluble compounds should have a minor role in physical structure of the plant matter. The decrease of a_s_ (40%) due to the ADF treatment (from FA to FB) suggests that hemicellulose plays a structural role in plant matter. An increase of a_s_ (24%) due to the H_2_SO_4_ treatment (from FB to FC) proves the presence of remaining cellulose molecules with a major structural role and so a partial efficiency in H_2_SO_4_ treatment.

## 4. Conclusions

In order to study plant cell wall composition, a non-conventional Van Soest version was carried out to produce samples with specific cell wall compounds. Thermal Gravimetric Analysis (TGA) of Van Soest Fractions cell wall showed successive decomposition and depolymerization of plant matter as treatment steps progressed. Results from TGA under O_2_ allowed:The emphasis of complexes between macromolecules in plant matter by their quantities and types.A better and detailed determination of hemp shiv chemical composition as highlighted in Figure 8: 2.5 ± 0.6% water, 4.4 ± 0.2% pectins, 42.6 ± 1.0% (Hemicellulose–Cellulose), 18.4 ± 1.6% (Cellulose–Hemicellulose), 29.0 ± 0.8% (Lignin–Cellulose) and 2.0 ± 0.4% linked lignin.

This method, involving the Van Soest method and TGA, promotes an estimation of true value in percentage estimation. Thus, TGA/Mass Spectrometry (MS) or TGA/Gas Chromatography (GC)/MS-FID on Van Soest fractions could be interesting to identify volatile by-product degradation and at the same time cell wall molecules. The continuation of this work with other global methods analysis like Selvedran, Prosky, Uppsala, etc. in order to give supplementary information about interactions of cell wall molecules.

This work also highlights physicochemical interactions between hemicellulose and water and will be used in the production processes of building materials especially binderless particleboard whose particles’ interactions are still unclear.

## Figures and Tables

**Figure 1 molecules-26-06334-f001:**
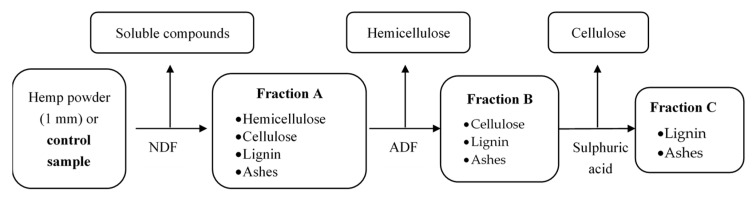
Chemical composition of Van Soest fractions.

**Figure 2 molecules-26-06334-f002:**
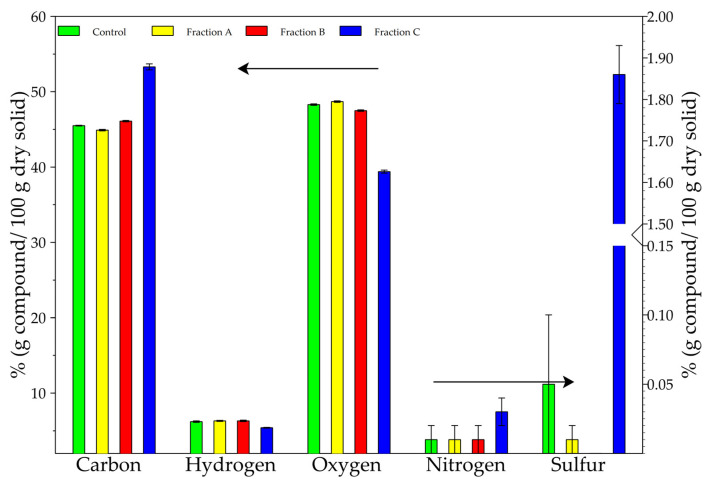
Comparative histogram bar chart of elementary analysis between control and fractions.

**Figure 3 molecules-26-06334-f003:**
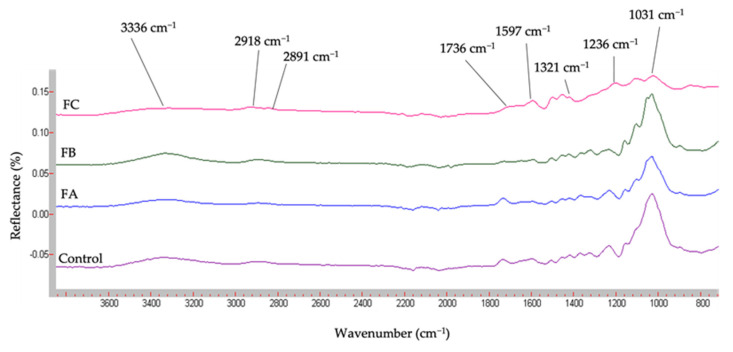
FTIR spectra of samples (Van Soest fractions and control).

**Figure 4 molecules-26-06334-f004:**
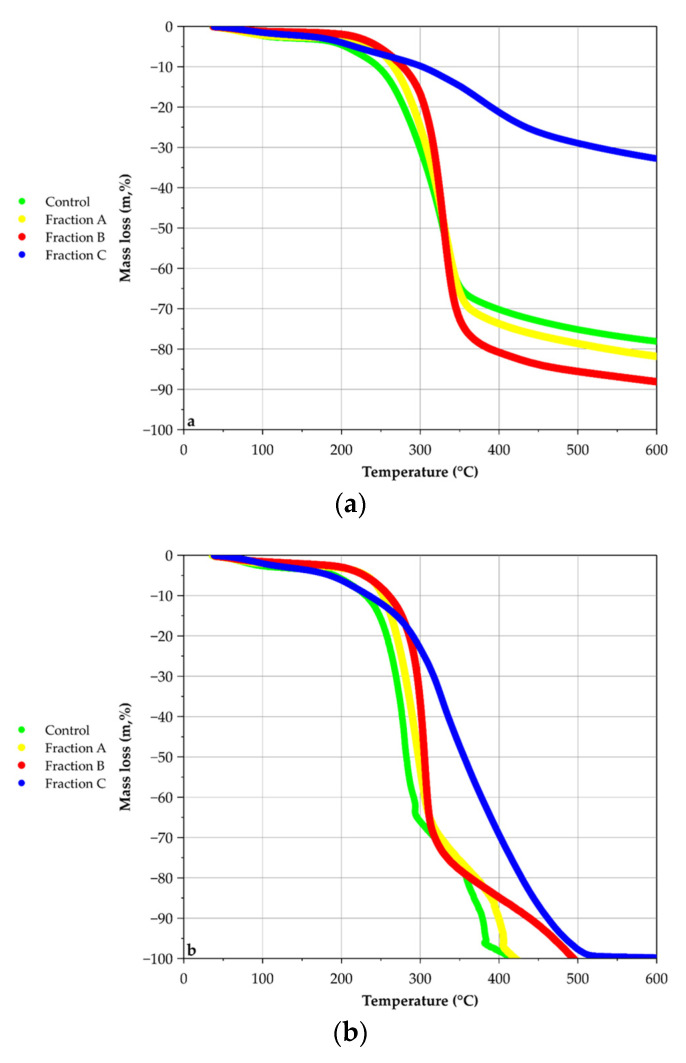
TGA hemp shiv fraction curves under Ar (**a**) and O_2_ (**b**).

**Figure 5 molecules-26-06334-f005:**
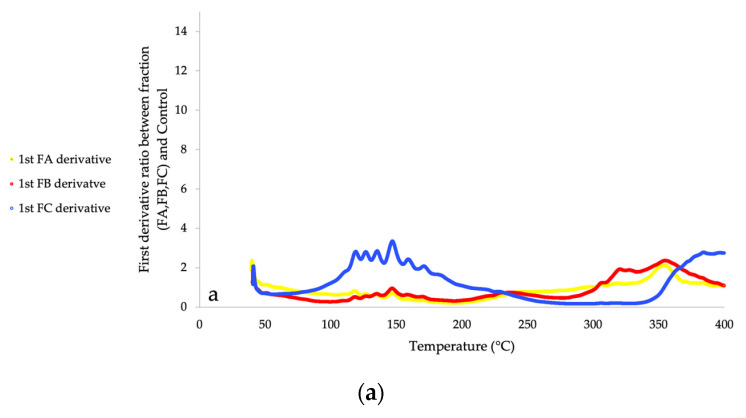
First derivatives ratio of Van Soest Fractions in compared to Control under argon (**a**) and oxygen (**b**).

**Figure 6 molecules-26-06334-f006:**
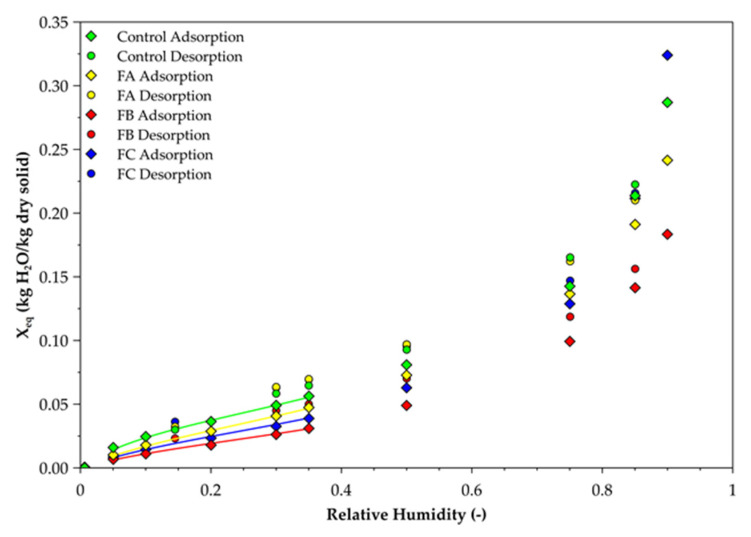
Moisture adsorption and desorption behaviour on Van Soest fraction at 23 °C. Number of repetitions = 2; SD max = 3.5%.

**Figure 7 molecules-26-06334-f007:**
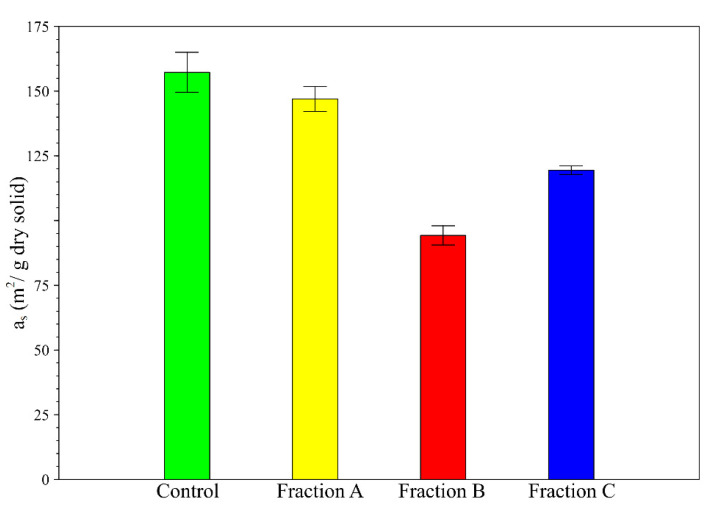
Specific area of Van Soest fraction (*a_s_*). Number of repetitions = 2.

**Figure 8 molecules-26-06334-f008:**
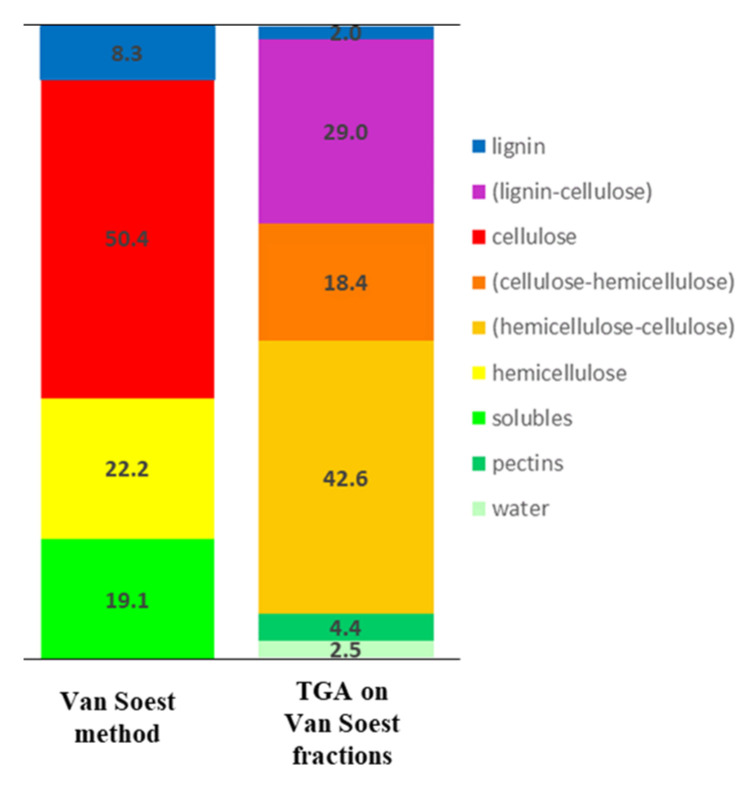
Comparative scheme of chemical composition results obtained with classic method Van Soest and the investigated method.

**Table 1 molecules-26-06334-t001:** Hemp shiv chemical composition according to the bibliography and expressed in percentage (% *w*/*w* dry mass).

References	Cellulose	Hemicellulose	Lignin	Soluble Compounds	Proteins	Ashes
Hussain et al., 2018 [13]	44.0	18.0–27.0	22.0–28.0	1.0–6.0	-	1.0–2.0
Vignon et al., 1995 [14]	44.0	18.0	28.0	5.0	3.0	2.0
Thomsen et al., 2005 [15]	48.0	21.0–25.0	17.0–19.0	-	-	-
Gandolfi et al., 2013 [16]	44.0	25.0	23.0	4.0		1.2
Garcia-Jaldon, 1995 [17]	48.0	12.0	28.0	7.0	3.0	2.0
Arufe et al., 2021 [11]	49.0	21.6	8.1	17.2		4.1
Arufe et al., 2021 [18]	46.1	21.5	8.5	21.1		2.8
Cappelletto et al., 2001 [19]	51.6	21.5	12.9	12.9		6.6
Godin et al., 2010 [20]	47.5	6.4	8.0	29.4		8.8
Viel et al., 2018 [10]	49.9	21.4	9.5	17.7		0.6

**Table 2 molecules-26-06334-t002:** Proposed determination protocol of onset (T_o_), peak (T_p_) and final (T_f_) characteristic temperatures of thermal transitions employing TGA data.

Nomenclature	1st Derivative	2nd Derivative
T_o_	dm/dT = 0	Inflexion point of d^2^m/dT^2^ vs. T
T_p_	LM	d^2^m/dT^2^ = 0
T_f_	dm/dT = 0	Inflexion point of d^2^m/dT^2^ vs. T

LM: Local Minimum.

**Table 3 molecules-26-06334-t003:** Cell wall hemp shiv composition by Van Soest method (% *w*/*w*). Number of repetitions = 6.

	Cellulose	Hemicellulose	Soluble Compounds	Lignin	Ashes
Dry basis (g ×/100 g dry mass)	49.0 ± 2.8	21.5 ± 1.7	18.6 ± 0.8	8.1 ± 0.6	2.8 ± 0.1
Organic mass (g ×/100 g organic mass)	50.4 ± 2.8	22.2 ± 1.8	19.1 ± 0.8	8.3 ± 0.6	-

**Table 4 molecules-26-06334-t004:** Percentage of C, H, N, S, O elements (% mg) in fractions of non-destructive Van Soest method and control. Number of repetitions = 3.

	% Carbon	% Hydrogen	% Nitrogen	% Sulfur	% Oxygen Calculated
Control	45.5 ± 0.1	6.2 ± 0.1	0.01 ± 0.01	0.05 ± 0.1	48.3 ± 0.1
FA	44.9 ± 0.1	6.3 ± 0.1	0.01 ± 0.01	0.01 ± 0.1	48.7 ± 0.1
FB	46.1 ± 0.1	6.3 ± 0.2	0.01 ± 0.01	0	47.5 ± 0.1
FC	53.3 ± 0.5	5.4 ± 0.1	0.03 ± 0.01	1.86 ± 0.1	39.4 ± 0.2

**Table 5 molecules-26-06334-t005:** Cell wall composition estimation (g/100g dry mass) under argon and oxygen with TGA ^1^. Number of repetitions = 3.

	Cell Wall Compounds	Control	FA	FB	FC
**Argon**	Water	2.7 ± 0.8 (40–140 °C)	2.0 ± 0.1 (45–130 °C)	1.2 ± 0.2 (40–130 °C)	2.3 ± 1.2 (40–130 °C)
	Pectins	3.5 ± 0.6 (140–220 °C)		83.8 ± 1.2 (180–510 °C)	4.0 ± 0.3 (130–230 °C)
	(Hemicellulose–Cellulose)	24.7 ± 4.6 (220–300 °C)	23.5 ± 2.1 (190–330 °C)		8.6 ± 1.1 (260–345 °C)
	(Lignin–Cellulose)	38.8 ± 6.4 (300–420 °C)	53.7 ± 3.4 (330–500 °C)		17.6 ± 1.0 (345–450 °C)
**Oxygen**	Water	2.5 ± 0.6 (40–120 °C)	2.0 ± 0.0 (45–150 °C)	1.5 ± 0.1 (40–115 °C)	3.0 ± 0.1 (45–140 °C)
	Pectins	4.4 ± 0.2 (150–220 °C)	74.6 ± 0.2 (160–360 °C)	80.3 ± 1.0 (180–385 °C)	5.2 ± 0.3 (145–230 °C)
	(Hemicellulose–Cellulose)	42.6 ± 1.0 (220–285 °C)			19.7 ± 10.7 (230–300 °C)
	(Cellulose–Hemicellulose)	18.4 ± 1.6 (285–310 °C)			26.4 ± 1.9 (300–355 °C)
	(Lignin–Cellulose)	29.0 ± 0.8 (310–400 °C)	22.7 ± 0.3 (360–430 °C)	16.6 ± 0.9 (410–500 °C)	48.3 ± 3.8 (350–540 °C)
	Lignin	2.0 ± 0.4 (400–430 °C)			

^1^ This cell wall composition estimation was based on Table 2.

## Data Availability

Not applicable.

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
