# Peer review of "Cell Wall Composition of Hemp Shiv Determined by Physical and Chemical Approaches"

_molecules, 2021, doi:10.3390/molecules26216334_

Round 1

Reviewer 1 Report

Minor remarks

Provide a blank space between quantity and unit, only in the case of percentage.

Greek symbols should be given in italics.

English language should be checked by a native speaker.

Major remarks

Please, avoid lumping the references in the manuscript. I suggest discussing each reference separately.

Also, older references can be excluded from the manuscript if they are not of key importance.

The conclusion is so long. I suggest retyping that and insert only important conclusions and give a major overview of further application.

Reviewer 2 Report

The manuscript " Cell wall composition of hemp shiv determined by physical and chemical approaches" deal with the study of cell wall composition of hemp shiv determined by physical and chemical approaches. In general, the manuscript is well written, the methodologies are precise and results well reported.

However, before to be considered for pubblications I suggest;

1) to improve the introduction part. The concept is clear, but the introduction can be slightly reorganized for a better understanding. For examples, in the first part the authors mention about application in construction but then nothing is reported about, so i believe is not necessary.

2) Table 1; Figure 1;Table 3;Table 5 should be resized 

3) in the figure caption, should be reported the number of measurement performed and the SD.

Reviewer 3 Report

General comments

The manuscript entitled “Cell wall composition of hemp shiv determined by physical and chemical approaches” represents a study regarding using high technology as a tool to clarify several assumptions already described in the literature in this regard.

Sometimes, the use of emergent technology leads to a discussion of ancients results as they were obtained by classic methodologies.

Comparing conventional techniques and the proposed approach, in the authors vision, which are the main advantage of this study? How different/similar they are, besides molecular interactions?

Abstract

In the presented form, this section is explaining the importance of this work (introducing it) instead of summing up the main goals/achievements in its regard. Please add some info about your results.

Introduction

Line 68, refs are missing

Line 73 – “O-acetyl-O-(…)”, When we are representing derivatives/moieties, the “O” should be presented in italic form. Please, revise the manuscript.

Results

Figure 4 – please correct “O2” for its correct molecular formula.

Conclusion

Line 467 – the “%” is missing for Cellulose-Hemicellulose amount.
